# A Comparison between the Molecularly Imprinted and Non-Molecularly Imprinted Cyclodextrin-Based Nanosponges for the Transdermal Delivery of Melatonin

**DOI:** 10.3390/polym15061543

**Published:** 2023-03-20

**Authors:** Gjylije Hoti, Riccardo Ferrero, Fabrizio Caldera, Francesco Trotta, Marta Corno, Stefano Pantaleone, Mohamed M. H. Desoky, Valentina Brunella

**Affiliations:** Department of Chemistry, University of Torino, Via P. Giuria 7, 10125 Torino, Italy

**Keywords:** cyclodextrin, molecularly imprinted nanosponges, non-moleculary imprinted nanosponges, melatonin, cream formulation, computational study

## Abstract

Melatonin is a neurohormone that ameliorates many health conditions when it is administered as a drug, but its drawbacks are its oral and intravenous fast release. To overcome the limitations associated with melatonin release, cyclodextrin-based nanosponges (CD-based NSs) can be used. Under their attractive properties, CD-based NSs are well-known to provide the sustained release of the drug. Green cyclodextrin (CD)-based molecularly imprinted nanosponges (MIP-NSs) are successfully synthesized by reacting β-Cyclodextrin (β-CD) or Methyl-β Cyclodextrin (M-βCD) with citric acid as a cross-linking agent at a 1:8 molar ratio, and melatonin is introduced as a template molecule. In addition, CD-based non-molecularly imprinted nanosponges (NIP-NSs) are synthesized following the same procedure as MIP-NSs without the presence of melatonin. The resulting polymers are characterized by CHNS-O Elemental, Fourier Transform Infrared Spectroscopy (FTIR), Thermogravimetric (TGA), Differential Scanning Calorimetry (DSC), Zeta Potential, and High-Performance Liquid Chromatography (HPLC-UV) analyses, etc. The encapsulation efficiencies are 60–90% for MIP-NSs and 20–40% for NIP-NSs, whereas melatonin loading capacities are 1–1.5% for MIP-NSs and 4–7% for NIP-NSs. A better-controlled drug release performance (pH = 7.4) for 24 h is displayed by the in vitro release study of MIP-NSs (30–50% released melatonin) than NIP-NSs (50–70% released melatonin) due to the different associations within the polymeric structure. Furthermore, a computational study, through the static simulations in the gas phase at a Geometry Frequency Non-covalent interactions (GFN2 level), is performed to support the inclusion complex between βCD and melatonin with the automatic energy exploration performed by Conformer-Rotamer Ensemble Sampling Tool (CREST). A total of 58% of the CD/melatonin interactions are dominated by weak forces. CD-based MIP-NSs and CD-based NIP-NSs are mixed with cream formulations for enhancing and sustaining the melatonin delivery into the skin. The efficiency of cream formulations is determined by stability, spreadability, viscosity, and pH. This development of a new skin formulation, based on an imprinting approach, will be of the utmost importance in future research at improving skin permeation through transdermal delivery, associated with narrow therapeutic windows or low bioavailability of drugs with various health benefits.

## 1. Introduction

The circadian rhythm is the 24 h internal clock in our brain that regulates the cycles of vitality and sleepiness by responding to light variations in our surroundings. The sleeplessness, due to lifestyle variables, affects the restoration of brain energy, and thus disrupts the circadian rhythm, causing metabolic disorders, cardiovascular, cancer diseases, etc. [1,2]. Melatonin (N-acetyl-5-methoxytryptamine) is an indoleamine neurohormone produced during the dark period by the pineal gland and is inhibited by light exposure. Melatonin is also produced in some extra sites than the pineal gland such as the liver, gut, ovary, retina, Harderian glands, lens, and bone marrow [3,4,5]. Melatonin biosynthesis begins from tryptamine and comprises serotonin formation. Serotonin is further N-acetylated and O-methylated to provide melatonin. Melatonin is known to treat sleeping disorders such as circadian rhythm sleep disorders, difficulty in staying and falling asleep, sleeping hurdles in blind people, jet-lag, and delayed sleep phase syndrome [1,6,7,8]. In addition to its effect on controlling sleeping, melatonin can also ameliorate various other health conditions such as slowing down the progression of Alzheimer’s disease, other neurodegenerative disorders, and protecting the organism from carcinogenesis, etc. [3,9,10,11,12]. Notwithstanding the aforementioned benefits, the drawbacks of melatonin are its oral and intravenous fast release. This may be related to the short plasma elimination half-life (~45 min) variable and low bioavailability of melatonin. It is well-established that the extensive hepatic first-pass metabolism can cause low bioavailability [13,14,15,16]. Melatonin is considered a poorly water-soluble substance (0.1 mg/mL), with a permeability characteristic (permeability coefficient in water, K_p_ = 7.20 ± 1.43 × 10^−4^) and a dissolution rate-limited absorption. Therefore, it can be classified as a class II category drug according to the Biopharmaceutics Classification System (poorly soluble and highly permeable) [6,12]. Consequently, the melatonin-sustained release formulations for oral, transdermal, intranasal, and transmucosal administrations have been considered [17]. The floating controlled-release melatonin-loaded chitosan microcapsules via ionic cross-linking of chitosan with sodium dioctyl sulfosuccinate (DOS) [18], the emulsion melting/cooling method to prepare melatonin-loaded microspheres using stearyl alcohol [19], sugar beads loaded with melatonin and coated with Aquacoat [14,20], β-Cylodextrin/melatonin inclusion complex in hydroxypropyl methyl cellulose (HPMC-K15M) and Carbopol 971 P matrix [21], melatonin starch microspheres [22], melatonin-loaded elastic liposomal formulation [23], melatonin-loaded ethanolic liposomes (ethosomes) [24], and lecithin/chitosan nanoparticles [25] are considered the adequate controlled-release delivery systems that have been developed to maximize the therapeutic effectiveness of melatonin. Enhancing drug delivery into the skin, through the development of transdermal drug delivery as an advance methodology, has been a major research focus for over half a century [26,27]. Melatonin delivery from the administration of transdermal, transmucosal, and oral-controlled release in human volunteers has been investigated. It is observed that transdermal melatonin delivery can ensure more benefits over oral administration because there is more melatonin deposited in the skin that can slowly leach from the skin, therefore, increasing the drug’s therapeutic profile during the removal of the sample formulation [28]. Transdermal administration is extensively used as it presents a longer duration of action in less dosing frequency, improved bioavailability, reduced side effects, and improved therapy due to the maintenance of uniform plasma levels over intravenous and oral routes [29]. Transdermal melatonin delivery can counteract the circadian wake drive during the day and can elevate the plasma melatonin, with the maximal concentration close to the physiological range for an extended duration [30], and can prevent first-pass metabolism [23,30]. The permeation of melatonin across the skin can be significantly ameliorated by using “passive” technologies such as chemical penetration enhancers, formulation excipients, and various nano and micro-delivery systems, hence reducing the barrier function of stratum corneum [26,28,31]. The nanosized platforms such as solid lipid nanoparticles, ethosomes, liposomes, niosomes, polymeric nanoparticles, and cyclodextrins (CDs) are considered for melatonin delivery to the skin [32]. The effect of ethanol, water, propylene glycol, and their binary and ternary mixtures on the melatonin permeation, using the response surface method and artificial neural networks, has been considered as the first step for developing a transdermal delivery system [31]. Polymers introduce a safe, effective, and constant drug delivery to the body due to the controlled drug release rate in the transdermal patch [33].

Cyclodextrins (CDs) are well-known to play a significant role in drug delivery, their designs are adjusted to the physicochemical properties of the activities [34]. CDs are low molecular weight (between 973 and 2163 Da) cyclic oligosaccharides that contain six (α-CD), seven (β-CD), eight (γ-CD), and a greater number of D-glucose units, joined through α-(1,4) glycosidic linkages. CDs are characterized by a typical toroidal cone shape with a lipophilic central cavity and a hydrophilic outer surface [35,36,37,38]. Owing to the peculiar central cavity, CDs tend to form inclusion complexes with organic molecules, non-polar drugs, or the nonpolar region of the molecule, consequently, improving their solubility, bioavailability, stability, controlled release, etc. Therefore, CDs are used in the pharmaceutical, gene therapy, cosmetic, agricultural, food, environment, packing, and textile industry, etc. CDs have also been utilized to optimize the dermal delivery of drugs considered for systemic and local use. These applications can be expanded through the modification of CDs. CDs can be modified due to the availability of multiple reactive hydroxyl groups [39,40,41,42,43,44]. Depending on the reaction and substituents, water-soluble and insoluble cyclodextrin-based polymers or cyclodextrin-based nanosponges (CD-based NSs) can be synthesized [45]. CD-based NSs are chemically cross-linked polymers that are synthesized by reacting the cyclodextrin (CD) unit with a suitable cross-linking agent [46,47].

CD-based NSs are efficient encapsulating agents to enhance the bioavailability and efficacy of drugs and deliver them with controlled kinetics through topical, oral, and parenteral routes [48,49]. The ability of CD-based NSs to entrap various molecules because of a highly porous nanomeric structure into the matrix has triggered extensive research and applications [50]. The historical development of CD-based NSs presents that these polymers have emerged over years [48,51]. Taking into account the chemical composition and properties, CD-based NSs can be classified into five consecutive generations. The fourth generation of CD-based NSs describes the molecularly imprinted polymers (MIPs) which can exhibit high selectivity, affinity, and specific molecule recognition towards specific molecules. Their synthesis is based on the formation of defined interactions between a functional monomer, cross-linking agent, and a template molecule [49].

In recent years, MIPs have been synthesized by molecular imprinting technology (MIT) that has attracted significant interest in research activity. Their synthesis, purification, and testing require precision and attention [52]. The imprinted polymeric carrier must exhibit high drug loading capacity, high release, and low toxicity to be used for therapeutic purposes. Due to the versatility of their polymeric scaffold, molecularly imprinted polymers (MIPs) are considered superior to other nano-systems [53]. The MIPs can offer a remedy when a large amount of melatonin is imprinted and cyclodextrin-based nanosponges (CD-based NSs) are used to prepare the nanocarrier for melatonin as they make inclusion complex [29]. A previous computational study [54], using static and dynamic simulations, provided a cost-effective methodology to investigate the inclusion complex between β-Cyclodextrin (β-CD) and melatonin. The theoretical results, in agreement with the experimental ones, are provided. This work presented a way of managing a structural model of the nanosponge. Further, the study in [55] investigated a bioactive functional fabric that was prepared through cotton fiber functionalization with melatonin-loaded β-CD carbonate non-molecularly imprinted nanosponges (NIP-NSs). The carbonate nanosponge was synthesized by reacting β-CD with 1-1′ carbonyldiimidazole (CDI) in N, N-dimethylformamide (DMF). The in vitro release presented zero-order kinetics, thus developing a novel biofunctional fabric that could control melatonin release through the skin. Another investigation [56] presented the capability of CD-based NSs, synthesized by reacting β-CD with pyromellitic dianhydride (PMDA) in dimethyl sulfoxide (DMSO), for binding capacities and specificities with glucose. This investigation confirmed the advantages of CD-based MIP-NSs over CD-based NIP-NSs regarding selectivity. Although CD-based NSs are investigated to encapsulate melatonin and various other drugs, their non-selectivity remains a challenging task of pharmacologic efficacy. CD-based molecularly imprinted nanosponges (MIP-NSs) are proposed as a promising drug delivery system for the protection and prolonged release of L-DOPA ((S)-2-amino-3-(3,4-dihydroxyphenyl) propanoic acid) [57].

However, the clinical applications of CD-based NSs as drug delivery systems have not yet been reached, since they are predominantly synthesized in organic solvents [58,59,60] and their safety issues invoke important remarks because of the presence of organic impurities that may cause cellular damage [61].

Therefore, this research results in the green synthesis of a series of cyclodextrin-based molecularly imprinted nanosponges (MIP-NSs) with melatonin as a template molecule. From [62], the nature of the interactions developed between MIPs and the template was the same as the ones developed between non-molecularly imprinted nanosponges (NIP-NSs), but the strength of these interactions was different. Accordingly, as a reference, cyclodextrin-based NIP-NSs without the melatonin were additionally synthesized following the same procedure as for MIP-NSs. The CD-based NSs were synthesized by reacting the β-CD or Methyl-β Cyclodextrin (M-βCD) with citric acid (CA) in deionized water. The MIP-NSs and NIP-NSs were mixed with a prepared cream formulation. An in vitro release study was carried out in a Franz Diffusion Cell system, and the release kinetics of melatonin from MIP-NSs and NIP-NSs cream formulations were compared with the one mixed with free melatonin (1 and 5% melatonin).

This work is an innovative study describing the water-based polymeric system for molecular imprinting to deliver melatonin through transdermal formulations.

## 2. Materials and Methods

### 2.1. Materials

β-cyclodextrin (β-CD) (M_w_ = 1134.98 g/mol) and methyl-β-cyclodextrin (M-βCD) (M_w_~1184 g/mol, with the degree of substitution DS of 0.5) were kindly supplied as gifts by Roquette Frères (Lestrem, France). β-CD and M-βCD were dried in an oven at a defined temperature up to constant weight before their usage to remove any traces of water.

Sodium hypophosphite monohydrate (NaPO_2_H_2_·H_2_O, ≥99%); potassium chloride (KCl, ≥99.5% (AT)); sodium chloride (NaCl, ACS, ISO, Reag. Ph Eur); glycerol (C_3_H_8_O_3_, 99+%); almond oil from Prunus dulcis; 2,6-Di-tert.-butyl-4-methylphenol ((BHT); [(CH_3_)_3_C]_2_C_6_H_2_ (CH_3_)OH, 99% (GC)); and Beeswax (bleached) were purchased from Sigma-Aldrich (Darmstadt, Germany).

The citric acid (C_6_H_8_O_7_, 99.9%) and acetonitrile (H_3_CCN, ≥99.9%, for HPLC-Isocratic grade) were purchased from VWR Chemicals BDH (Milan, Italy).

Melatonin (C_13_H_16_N_2_O_2_, 99+%), Alfa Aesar, was purchased from ThermoFisher (Kandel, Germany) GmbH.

Disodium hydrogen phosphate dodecahydrate (Na_2_HPO_4_·12H_2_O, 99%) and potassium phosphate monobasic (KH_2_PO_4_, 98%) were purchased from Italia Carlo Erba S. p. a., (Milan, Italy).

Ortho-phosphoric acid (H_3_PO_4_, 84–85%), 1-hexadecanol (C_16_H_34_O, ~99% (GC)), or cetyl alcohol, were purchased from Fluka Chemie GmbH, Buchs, Switzerland.

Stearate PEG-100 was donated by Evonik Industries AG (Essen, Germany).

All chemicals were analytical-grade commercial products and were used as received. Deionized water, and water purified by reverse osmosis (MilliQ water, Millipore, Burlington, MA, USA) with a resistivity above 18.2 MΩcm^−1^, dispensed through a 0.22 μm membrane filter, were used throughout the studies.

### 2.2. Methods

#### 2.2.1. Synthesis of Cyclodextrin-Based Molecularly Imprinted and Non-Molecularly Imprinted Nanosponges (MIP-NSs and NIP-NSs)

Cyclodextrin-based molecularly imprinted nanosponges (labelled as MIP-NSs) were successfully synthesized following the procedure in [63,64] with some modifications. β-Cyclodextrin (β-CD) or Methyl-β-cyclodextrin (M-βCD) which were used as functional monomers, citric acid as the cross-linking agent, and sodium hypophosphite monohydrate as the catalyst, were mixed in 10 mL deionized water as the solvent. The millimoles (mmol) are presented in Table 1. The polymers were synthesized with 8 moles of citric acid per mol of β-CD or M-βCD. Alongside these, melatonin (10%, 20%, and 50%, with respect to the mol of β-CD or M-βCD) was used as a template molecule. The reaction was carried out in a vacuum oven at specific temperatures (140 °C, and 100 °C) and times (mainly 1 h at 140 °C, and >15 h at 100 °C). The solidified mass was broken up and manually ground in a mortar or milled using the planetary ball mill. Following the same protocol and conditions, cyclodextrin-based non-molecularly imprinted nanosponges (labelled as NIP-NSs), in the absence of melatonin as the imprinting molecule, were synthesized.

#### 2.2.2. Fourier Transform Infrared Spectroscopy (FTIR) Analysis

The synthesized MIP-NSs and NIP-NSs were characterized utilizing Fourier Transform Infrared Spectroscopy (FTIR), using a Perkin Elmer Spectrum Spotlight 100 FTIR spectrophotometer equipped with Spectrum software. The FTIR spectra were gained in the spectral range of 4000–650 cm^−1^, at a spectral resolution of 4 cm^−1^ and a sample/background scan number of 8. FTIR spectra were obtained using a versatile Attenuated Total Reflectance mode (FTIR-ATR) sampling accessory with a diamond crystal plate. The FTIR-ATR measurements were performed on dried samples.

#### 2.2.3. Computational Details

All the simulations were carried out with the xTB code (extended Tight Binding) v. 6.4.0 at the semi-empirical level of theory xTB-GFN2 [65,66]. The lowest energy structures were obtained by a conformational search, employing the submodule Conformer-Rotamer Ensemble Sampling Tool (CREST) [67] as already applied in previous work and as documented in the Appendix A. Structures obtained by CREST were then re-optimized in the gas phase at a GFN2 level using tighter thresholds. MOLDRAW [68] and VMD [69] graphical codes were employed for the molecular modelling of inclusion complexes.

#### 2.2.4. Thermogravimetric Analysis (TGA)

The thermal stability of the synthesized MIP-NSs and NIP-NSs was studied by thermogravimetric analysis (TGA). TGA was carried out using a TA Instrument Thermogravimetric Analyzer (TGA) Q500 from room temperature up to 800 °C, under nitrogen (N_2_) flow, and with a heating ramp rate of 10 °C/min. The gas flows applied in the balance and furnace section were 40 mL/min and 60 mL/min. About 10 mg of the sample was weighed in an aluminum pan for analysis.

#### 2.2.5. Differential Scanning Calorimetry (DSC)

Differential Scanning Calorimetry (DSC) measurements were carried out using a TA instrument Q200 DSC (New Castle, DE, USA) on a 2–3 mg sample under a nitrogen flow of 50 mL/min. The empty aluminum pan was used as a reference standard. A heating rate of 10 °C/min was employed in the 20–130 °C temperature range.

#### 2.2.6. CHNS-O Elemental Analysis

CHNS-O elemental analysis (EA) was carried out to quantify accurate and reproducible carbon, hydrogen, nitrogen, and sulfur content in the synthesized polymers using a CHNS-O Analyser (Thermo Fisher Scientific FlashEA 1112 series; Waltham, MA, USA), equipped with the Eager Xperience software (for Windows XP) and a MAS 200R Auto Sampler. 2,5-Bis (5-tert-butyl-2-benzo-oxazol-2-yl) thiophene (BBOT) was used as an external standard for the calibration of the system. About 2.5 mg of each sample and an, approximately, equal quantity of V_2_O_5_ as a catalyst were placed in a tin container. The CHNS-O analyses were performed in duplicate. The nitrogen signal enabled the estimation of the melatonin presence in the MIP-NSs.

#### 2.2.7. Scanning Electron Microscopy (SEM)

The morphology of the synthesized MIP-NSs and NIP-NSs was studied using scanning electron microscopy (SEM). The imaging was conducted with a Zeiss EVO 50 (Oberkochen, Germany) using secondary electrons and a 10 kV accelerating voltage. The samples were placed on the aluminum stub, with the help of a bio-adhesive carbon tape. Before SEM analysis, the polymers were ion-coated with gold using a Baltec SCD 050 sputter coater (Pfäffikon, Switzerland) for 40 s under vacuum at 60 mA.

#### 2.2.8. Zeta Potential

Zeta potential was measured using a Malvern Zetasizer Nano ZS (Brookhaven Instruments Corporation, Holtsville, NY, USA). Approximately 1 mg of the synthesized sample was dispersed in 1 mL of MilliQ water and shaken for a certain time. During zeta potential analysis at 25 °C, the prepared dispersions were placed in the electrophoretic cell and the particles moved to the electrode that had an opposite charge to them upon the application of an electric field (15 V/cm). Three measurements were carried out, and their average was presented.

#### 2.2.9. Water Absorption Capacity (WAC)

The kinetics of the swelling of synthesized MIP-NSs and NIP-NSs was studied by following their increase in weight when immersed in water. The swelling measurements were performed by immersing 0.5 g of dry powder in deionized water (in 15 mL test tubes) and blending them, in the beginning, using a Vortex Mixer. The test tubes were sealed and maintained at room temperature. After 2 h, the mixtures were centrifuged to obtain a layer of water-bound material and free, unabsorbed water. After removing the supernatant, the residual amount of free water was blotted off using tissue paper, and the weight was recorded. The WAC measurements were performed in duplicate. The water absorption capacity (%WAC) was calculated using the following Equation (1):(1)$$WAC\left(\%\right)=\frac{\left(m_{t}-m_{o}\right)}{m_{o}}\times 100$$
where *m_t_* is the weight of the swollen sample at time *t*, and *m_o_* is the initial weight of the dry sample [45].

#### 2.2.10. Melatonin Stability

The storage stability of melatonin was investigated under respective mediums such as phosphate-buffered solutions of pH = 6.04, 6.82, 7.00, and 7.40.

0.5 mg of melatonin was solubilized in 5 mL of each medium. Buffer solutions were prepared as follows: pH = 6.04 (1.54 g of Na_2_HPO_4_·12H_2_O, and 5.04 g of KH_2_PO_4_, 8.58 mL of Na_2_HPO_4_·12H_2_O, and 74.2 mL of KH_2_PO_4_); pH = 6.82 (4.10 g of Na_2_HPO_4_·12H_2_O, and 2.14 g of KH_2_PO_4_, 22.8 mL of Na_2_HPO_4_·12H_2_O, and 31.4 mL of KH_2_PO_4_); pH = 7.00 (4.62 g of Na_2_HPO_4_·12H_2_O, and 1.52 g of KH_2_PO_4_, 25.8 mL of Na_2_HPO_4_·12H_2_O, and 22.4 mL of KH_2_PO_4_); and pH = 7.40 (5.40 g of Na_2_HPO_4_·12H_2_O, and 0.70 g of KH_2_PO_4_, 30 mL of Na_2_HPO_4_·12H_2_O, and 10.3 mL of KH_2_PO_4_). The melatonin solutions were analyzed at different interval times for a long period (0 h, 3 h, 20 h, 24 h, 43 h, 48 h, 7 days, 2 weeks and 3 weeks) with a UV-Vis spectrophotometer.

Further, the thermal stability of melatonin at various temperatures (60 °C, 80 °C, 100 °C) for 5 h was also investigated and analyzed by TGA.

#### 2.2.11. UV-Vis Spectrophotometer

UV-Vis absorption spectra were obtained with a Hewlett Packard UV-Vis spectrophotometer model 845-3. A 10 mm path-length rectangular quartz cell (Hellma) was employed in the 200–400 nm spectral range. The concentration of the samples was adjusted so that the extinction values did not exceed A = 1.

A value of 223 nm was chosen as the λ_max_ of melatonin.

#### 2.2.12. Loading of Melatonin into NIP-NSs

Melatonin-loaded NIP-NSs was prepared by dissolving 20% (0.86 mmol with respect to the mmol of β-CD, and 0.82 mmol with respect to the mmol of—β-CD) in 10 mL 4:6 (*v*:*v* ratio) EtOH/H_2_O solution. To this solution, 0.5 g of the NIP-NSs was added, and the dispersion was mildly stirred for 24 h at room temperature. The melatonin-loaded NIP-NSs was recovered through centrifugation and freeze-dried.

#### 2.2.13. The Cream Formulation for the Delivery of Melatonin

The cream formulation comprises the oil phase made of almond oil, stearate PEG-100, beeswax, cetyl alcohol, and 2, 6-Di-tert.-butyl-4-methylphenol (BHT), which were placed in a 150-mL beaker. The beaker was directly heated on a hot plate at 75 °C until all the ingredients melted. While this mixture was kept warm, the aqueous phase, made of glycerol, and deionized water was further prepared in a 150-mL beaker and heated on a hot plate at 75 °C.

After the aqueous phase reached 75 °C, it was slowly added into the oil phase, under constant stirring, with the aid of an overhead stirrer (Model RW 20 digital, IKA, Staufen, Germany), until a smooth and uniform paste was reached. The quantities of the ingredients are presented in Table 2.

#### 2.2.14. Incorporation of MIP-NSs and NIP-NSs in a Cream Formulation

Due to their dried texture [70], the MIP-NSs or NIP-NSs are deemed incapable to be directly applied over the skin. Therefore, 0.5 g of MIP-NSs or NIP-NSs was incorporated under continuous stirring in 3 g of cream formulation. The formed homogeneous formulations were allowed to stand for 15 min without stirring to expel the trapped air, and were then stored in tightly closed screw-capped bottles at 5 °C.

#### 2.2.15. Evaluation of the Pharmaceuticals Parameters of Cream Formulations

##### Centrifuge Test

For pre-screening purposes, cream formulations underwent a centrifugation test, using the 4232-D Benchtop + 5228 RPM Tube Rotor (A.L.C. International S. r. l., Milan, Italy) centrifuge at a certain weight and 4000 RPM for the duration of 30, 60, and 90 min to evaluate syneresis, precipitation, or phase separation. At defined time intervals, the formulations were controlled to observe if there was any change [71].

##### Determination of pH

The pH of the cream formulations was measured with the aid of a digital pH meter (Orion model 420 A). A total of 0.10 g of MIP-NSs cream was uniformly dispersed in 25 mL of deionized water and kept for 2 h at room temperature. The pH of dispersions was measured at 25 °C [72].

##### Rheological Measurements and Determination of Viscosity

The rheological characterization of prepared creams was evaluated using the Rheometer TA Instruments Discovery HR 1 by following the procedure described in the literature with some modifications [73,74,75]. The rheograms’ shear stresses (τ), as a function of shear rate (γ’), were recorded in the Trios 5.0.0.44608 software, coupled with the rheometer. The instrument was equipped with 20 mm diameter stainless steel plate geometry, Peltier plate temperature control, and an air compressor unit. The viscosity of prepared creams, a rheological parameter, was measured as a function of shear rate (0.1–120 s^−1^). This enabled the study of the effect of MIP-NSs on cream viscosity. The temperature was maintained at a constant 25 °C by a water bath circulator. The sample was placed between the upper parallel plate and stationary surface with a 1 mm gap size. After sample loading, the gap was closed, and the sample edge was carefully trimmed with a spatula to maintain the proper surface shape during the measurements and avoid errors. Measurements were carried out in steady flow using a protocol with three steps such as (1) Conditioning-sampling, where the sample was equilibrated for 2 min before the experiments to allow the relaxation of the whole structure. A delay of 2 min was applied to measure the initial structure level of the samples before shearing to eliminate any disturbance created by the measuring geometry; (2) Flow-Ramp, increasing the linear shear rate from 0.1 to 100 s^−1^ in 2 min; and (3) Flow-Peak hold, maintaining a constant shear rate at 100 s^−1^ over 2 min. To ensure the reproducible state of the samples, the measurements were accomplished in triplicate recording their average. The data obtained were expressed as mean values ± SD.

##### Determination of Spreadability

The spreadability of the cream was determined using the parallel plate method following the procedure in the literature with slight modifications [76]. A total of 0.35 g of cream was placed at the center of the glass plate. Furthermore, above it, another glass plate was positioned concentrically and was allowed to rest for ~30 min. The cream was spread in a circle, and thus its diameter was determined. Spreadability was determined using the following Equation (2):(2)$$S=\frac{M\times L}{T}$$
where *S* is the spreadability, *M* is the weight tied to the upper side, *L* is the length of the glass slide, and *T* is the time taken to separate the slide from each other. Spreadability was recorded in g·cm/s.

#### 2.2.16. Evaluation of Melatonin-Loaded Efficiency

The extraction of melatonin was accomplished by stirring 0.1 g of nanosponge in 5 mL of H_2_O: ACN (70:30 *v*:*v*, pH = 3) solution. After 20 min, the dispersion was centrifuged at 6000 RPM for 10 min, and the supernatant was recovered as the first extract and then replaced with 5 mL of fresh H_2_O:ACN (70:30 *v*:*v*) solution. The extraction was performed three times. A total of 1.5 mL of each extract was filtered through a membrane filter (0.2 μm PTFE, Millipore, 13 mm Syringe Filter) and was analyzed with HPLC-UV (λ_Mel_ = 223 nm). Melatonin was quantified against an external calibration curve with standards (1, 2, 5, 10, 20, 50, 70, 100 μg/mL) prepared by serial dilution with a mobile phase of a 1000 μg/mL stock solution. The calibration curve (R^2^ = 0.9999) was used to quantify the melatonin extracts. The measurements were carried out in triplicate. Loading capacity (LC) was defined as the mass of encapsulated melatonin (*m_en_*), divided by the total mass of nanosponge (NS) (*m_tot_*), as presented in Equation (3):(3)$$\mathrm{LC}\;(\%)=\frac{m_{en}}{m_{tot}}\times 100$$

The encapsulation efficiency (EE) was defined as the mass of encapsulated melatonin (*m_en_*) over its initial mass (*m_in_*) during the loading step, as presented in Equation (4) [77]:(4)$$\mathrm{EE}\;(\%)=\frac{m_{en}}{m_{in}}\times 100$$

#### 2.2.17. In Vitro Release Study

Melatonin-release profiles were evaluated in a phosphate-buffered solution (PBS; 8 g of NaCl, 0.2 g of KCl, 3.63 g of Na_2_HPO_4_·12H_2_O, 0.24 g of KH_2_PO_4_ dissolved in 1 L water), with a pH of 7.4 as the receiving phase.

In vitro release tests of melatonin from cream formulations were conducted on static vertical Franz Diffusion Cells (PermeGear, Hellertown, PA, USA) [55]. The Franz Cells were constituted by an upper donor compartment and a lower receptor compartment (volume 12 mL) that were separated by a dialysis membrane with a contact surface of 1 cm^2^ and a thickness of 45 μm. The membrane was immersed overnight in a pH 7.37 phosphate-buffered solution. The sample was placed on top of the membrane in the donor compartment. The receptor compartment was filled with 8 mL of pH 7.37 phosphate-buffered solution and was kept at 37 °C by a heating jacket under constant stirring (500 rpm). At fixed intervals (30 min, 1 h, 2 h, 3 h, 4 h, 5 h, 6 h, 7 h, 23 h, and 24 h), the entire medium (8 mL) from the receptor compartment was withdrawn and replaced with the same volume of fresh medium. Each collected medium was filtered through a membrane filter (0.2 μm polytetrafluoroethylene (PTFE), Millipore, 13 mm Syringe Filter), and the amount of released drug was assessed with HPLC-UV. A standard calibration curve of melatonin at λ = 223 nm was obtained (R^2^ = 0.9995). The in vitro release tests were performed in triplicate.

#### 2.2.18. High-Performance Liquid Chromatography (HPLC) Analysis

High-Performance Liquid Chromatography (HPLC) analysis was carried out at room temperature using a PerkinElmer Brownlee Analytical C18 chromatographic column (250 mm × 4.6 mm, particle size 5 μm) connected to a PerkinElmer HPLC system, comprising a Flexar pump working at a flow rate of 1 mL/min and a Flexar UV–vis detector set at 223 nm (λ_max_ of melatonin). The mobile phase was prepared by mixing water (H_2_O) and acetonitrile (ACN) (H_2_O:ACN; 70:30 *v*:*v*), at a pH = 3 (adjusted with H_3_PO_4_, 84–85%), and the elution was isocratic. The total run time was set to 10 min, while the retention time of melatonin was observed at 8 min.

## 3. Results and Discussion

### 3.1. Synthesis of Cyclodextrin-Based Molecularly Imprinted and Non-Molecularly Imprinted Nanosponges (MIP-NSs and NIP-NSs)

The polymerization reaction led to a successful synthesis of cyclodextrin-based molecularly imprinted and non-molecularly imprinted nanosponges (MIP-NSs and NIP-NSs). Appendix A presents the mechanism of the formation of citrate nanosponges (NSs) [78,79,80,81,82]. Citric acid is a non-toxic cross-linking agent used to cross-link starch and, therefore, was also chosen to cross-link cyclodextrin (CD). The resulting products were further characterized to understand the interaction between melatonin and citrate MIP-NSs. Figure 1 presents the schematic representation of the simplified synthesis of a molecularly imprinted polymer (MIP). The proper selection of the functional monomer, template, solvent, and cross-linking agent are some of the most important aspects to be considered while synthesizing a MIP [53]. The synthesized MIP, as presented in Figure 1, could recognize the specific target molecule. This is because the MIP was prepared in the presence of the target molecule, thus generating complementary and specific binding sites within the polymer matrix [83]. In addition, to understand if improved melatonin loading and release, NIP-NSs were synthesized following the same procedure as MIP-NSs without adding the melatonin. The melatonin was loaded into NIP-NSs, following the procedure described in Section 2.2.12. This study presents a green material, based on an environmentally safe chemical reaction with water as a solvent and its easy production, as a candidate with huge potential to deliver melatonin. Therefore, it will become a superior alternative to the classical synthesis of CD-based MIP-NSs [57]. From the literature, in the case of large molecules, approximately 25% of the permanence original template was observed since the 100% removal of the template, due to highly cross-linked regions, was, hardly, achieved. This can be related to the inadequate solubility of the template in the solvent to interrupt the interactions with the imprinted cavity or to weak attainability of solvent to highly cross-linked areas [84]. According to the literature, the rigid structure of the MIP remains unchanged after template removal due to the strong binding of the functional monomer, both through the cross-linking agent and around the template [57]. Therefore, the MIP-NSs without the template removal are investigated in this study.

### 3.2. Melatonin Stability

The storage stability of melatonin buffer solutions in pH 6.04, 6.82, 7.00, and 7.40, stored at room temperature, was investigated by a UV-Vis spectrophotometer. It can be observed from Appendix A that the UV-Vis spectra were similar, and the melatonin was stable at various pH and interval times. The characteristic absorption peak of melatonin was recorded at ~280 nm. Appendix A confirmed the thermal stability of melatonin at different temperatures (60 °C, 80 °C, 100 °C) for 5 h, since no degradation was observed under the influence of temperature. These results aid in developing greener and more efficient alternatives for the synthesis of MIP-NSs, incorporating melatonin as a template molecule.

### 3.3. Physicochemical Characterization

The thermal analysis of MIP-NSs and NIP-NSs was analyzed by the TGA (Figure 2a,b). The first percentage of mass loss was related to the loss of water associated with the polymer (as detailed in Appendix A), observed at temperatures up to 100 °C. The maximum degradation processes of the βCD and M-βCD occurred at around 250 to 400 °C. The thermogram of MIP-NSs and NIP-NSs presented two degradation steps. The first one could be related to the degradation of the drug. The second one, or the main weight loss step from 200 to 400 °C, was related to the maximum degradation processes of the cross-linked structure of NSs. The two-stage decomposition profile in MIP-NSs and NIP-NSs was because of the ester groups, which were formed during the reaction of βCD or M-βCD with citric acid [6,82]. The TGA of MIP-NSs presented a similar maximum degradation behavior to NIPs, indicating their good thermal stability.

The changes in the peaks of the TGA curves of β-CD or M-βCD from the MIP-NSs or NIP-NSs were observed due to changes in chemical structure, which led to the formation of polymeric materials.

The DSC thermograms (Figure 3a,b) presented the formation of MIP-NSs and NIP-NSs as a result of the disappearance of melatonin peak that had a sharp endothermic transition at 120 °C (Appendix A), corresponding to its melting point. Whereas the large endothermic peak associated with water loss, between 50 and 100 °C, was observed for βCD-based MIP-NSs or βCD-based NIP-NSs and M-βCD-based MIP-NSs or M-βCD-NIP-NSs. The disappearance of the melatonin endothermic peak indicated the molecular association of the melatonin with the structure of synthesized polymers [85].

The formation of MIP-NSs and NIP-NSs was supported by FTIR measurements. The FTIR-ATR spectrum presented the prominent absorption of β-CD or M-βCD: 3342 cm^−1^ (O-H stretching vibrations), 2923 cm^−1^ (C-H stretching vibrations), 1254 cm^−1^, and 1020 cm^−1^ (C-O stretch). The cross-linking of citric acid with β-CD or M-βCD was presented by a characteristic band at 1720 cm^−1^, assigned to the C=O stretching vibrations of the cross-linking group. Whereas the absorption at 1624 cm^−1^ was due to the carboxylic acid moieties. All the synthesized samples presented comparable spectra with similar bands at characteristic wavenumbers and similar relative intensities, which displayed no interaction of melatonin with MIP-NSs or NIP-NSs (Appendix A). The enlargement of the absorption band in the range 3600–2950 cm^−1^ was attributed to the presence of the OH groups of carboxylic groups in the MIP-NSs and NIP-NSs. The knowledge of the polymer-drug interactions was more essential for the development of the delivery systems. Therefore, to further investigate these interactions, the concurrent adoption of computational methods was highly required. To achieve a better understanding and a more accurate assignment of the experimental IR bands, the concurrent adoption of computational methods was highly required. To this end, the IR spectrum was simulated with a semiempirical method at the GFN2 level of theory, employing the most stable structure of the melatonin/β-CD complex, as obtained from the CREST procedure [67] (Figure 4 and Appendix A). The calculated binding energy was equal to 33.13 kcal/mol, whose contribution given by dispersive forces was 19.42 kcal/mol, thus showing that 58.6% of the melatonin/cyclodextrin interactions were dominated by weak forces (as detailed in a previous study [56]).

A detailed comparison (Figure 5) between the characteristic computed peaks and the experimentally observed ones was essential to derive more solid information regarding the host–guest inclusion complex. Between 3800 and 3400 cm^−1^, the very broad peak of the hydroxyl group could be found, covering and including the N-H stretching signal (at 3300 cm^−1^ usually), typical of the amide group of melatonin. In the range 3050 to 2900 cm^−1^, there were the C-H stretching signals, while there was a single, very intense peak representing the stretching of the carbonyl groups of the melatonin amide group at 1730 cm^−1^. The higher intensity of this peak in the experimental spectrum could be explained by the presence of the carboxyl group of citric acid used as a cross-linking agent to synthesize CD-based nanosponges (CD-based NSs) in an 8:1 molar ratio concerning β-CD. However, a broader shoulder at 1650 cm^−1^ emerged from the citrate signal of MIP-NSs (green spectrum), in good accordance with the computed ν(CO) amide peak of melatonin. Therefore, it could be concluded that the main peak was the ester CO of citrate, while the peak tailing was due to the CO of melatonin. The digital fingerprint zone showed good superposition among the experimental and computational spectra as well, even for what concerns the relative intensities.

Although the computed IR spectrum was based on the single inclusion complex melatonin and β-CD, good agreement between the experimental and computational ones seemed to suggest the proper insertion of the guest molecule (melatonin) inside the cavity, confirming the creation of a multiple series of inclusion complexes, to further lead to the formation of nanosponges (NSs).

CD-based NSs are very delicate sponge-like cross-linked structures. The surface structure of cross-linked MIP-NSs and NIP-NSs was further morphologically studied by the SEM technique (Figure 6). SEM images at 1500 magnification confirmed the heterogeneous polymer surface [43]. The particles of the MIP-NSs and NIP-NSs presented an irregular form and were larger than 10 μm in diameter. There were no significant differences observed regarding the size and morphology of MIP-NSs and NIP-NSs. However, despite their irregular shape, the particles exhibited a good release rate of melatonin, as was indicated by the in vitro release profile. According to the literature, the morphology of the particles was important for the controlled release of the encapsulated materials, since it affects the interactions of the particle surface with the medium [86].

### 3.4. Water Absorption Capacity (WAC) and Zeta Potential (ζ-Potential)

The capacity of NIP-NSs to absorb water, as a function of the presence of functional groups in the polymer matrix, was determined. As the network of NIP-NSs bears hydroxyl and carboxylic acid groups, there was a high affinity for water molecules [45]. The water absorption capacity (WAC) of NIP-NSs was compared with the one of MIP-NSs. The WAC of NIP M-βCD: CA NSs (237%) was higher than the NIP βCD: CA NSs (126%), as it is presented in Table 3. This could be explained by the fact that the native cyclodextrin (CD) is 10-fold less soluble than the M-βCD: CA, as it was presented in the literature [87,88]. The presence of methyl groups could affect the structure of the nanosponge, and thus the water absorption capacity also. The water absorption or swelling capacity, a fundamental property of the cross-linked polymers, influence the drug release rates. The release of a drug incorporated into the polymer matrix is complex. Both the diffusion rate of the drug through the swollen polymer and the rate of water absorbed affect the overall drug release [89,90,91]. It can also be observed that the presence of melatonin reduces water absorption (Table 3). This is related to what has already been observed in the literature [6], in that melatonin is partially water-soluble, and, therefore, leads to insufficient absorption. Moreover, the increasing of the WAC could enhance the free space by releasing more drugs [92]. The design of MIPs, for drug delivery or analytical applications, requires a specific synthetic approach. Based on analytical purposes, a 1:8 molar ratio between β-CD or M-βCD and citric acid in a high quantity as a cross-linking agent was chosen to provide a remarkable binding of the template molecule through its rigid cavities. However, for drug delivery, the lower cross-linking ratio and a high WAC were preferable to tune release kinetics. NIP-NSs formed opalescent colloidal dispersions in MilliQ water. The Zeta potential (ζ-potential) of the MIP-NSs and NIP-NSs was found to be negatively charged, ranging from −24 to −33 mV (Table 3). The values were sufficiently high enough to confirm the stability of the MIP-NSs and NIP-NSs dispersions, meaning that they will not undergo aggregation over time.

### 3.5. Evaluation of Melatonin-Loaded Efficiency

The encapsulation and release behavior of the melatonin loaded into MIP-NSs and NIP-NSs was investigated and quantified by HPLC-UV. Melatonin is poorly water-soluble, but it is soluble in ethanol (20 mg/mL). In this study, the ethanol was mixed with water (4:6; Ethanol:Water; *v*:*v* ratio) to affect the solubilization capacity of the melatonin. The ethanol molecules were completely miscible with the water molecules, thus decreasing the dielectric constant. The lower dielectricity of the ethanol/water mixtures led to the solubility of non-soluble drugs compared to pure water [55,93,94]. Ethanol is a well-known penetration enhancer that can affect the intercellular region of the stratum corneum. The high concentration of ethanol in liposomes formed the ethosomes, which presented a high encapsulation efficiency for various lipophilic drugs and thus can be effectively considered for transdermal drug delivery systems [24]. The melatonin loaded into βCD-based NIP-NSs (Table 4) was the highest, with a loading capacity of 6.51%, whereas its encapsulation efficiency was 39.11%. The NIP-NSs (LC; 4–7%) presented a higher drug-loading capacity than the MIP-NSs (LC; 1–1.5%). Whereas the encapsulation efficiency was higher for MIP-NSs (60–90%) than the NIP-NSs (20–40%). The loading capacity of melatonin into NIP-NSs was comparable with what had been already observed in a previous study. The loading capacity of melatonin was estimated at 8 wt% [55]. Meanwhile, an in vitro release study, utilizing a Franz Diffusion Cell, was performed.

The influence of melatonin concentration (10%, 20%, and 50% melatonin with the respect to the mmol of β-CD and M-βCD), in loading capacity, was investigated. With the increase in melatonin quantity in MIP-NSs and NIP-NSs, the loading capacity and encapsulation efficiency significantly increased. CHNS-O elemental analysis confirmed (as presented in Table 5) the presence of nitrogen due to the presence of melatonin in the MIP-NSs. When the amount of melatonin in the MIP-NSs was increased, the percentage of nitrogen (CHNS-O elemental analysis) or loading capacity (HPLC-UV) was increased. To evaluate the imprinting effect, the selectivity of MIP-NSs was further investigated with the in vitro drug release study.

### 3.6. The Release Study In Vitro

The possibility to translate the MIP-NSs investigated in this study, from academia to the commercial stage, was considered by developing an appropriate topical formulation to enhance the permeation of melatonin into the skin. To be further utilized for the improvement of the transdermal delivery of melatonin, the MIP-NSs and NIP-NSs were mixed in a cream formulation. The release profiles of melatonin from cream formulations, using a Vertical Franz Diffusion Cell, are shown in Figure 7. There were various observed in vitro release profiles of melatonin. The best performances were obtained for the βCD-based MIP-NSs, employing 10% of melatonin, and for the M-βCD-based MIP-NSs, employing 20% of melatonin. The range percentage of released melatonin for 24 h, with the different formulations, was 33–48% for βCD-based MIP-NSs and 36–52% for M-βCD-based MIP-NSs. Whereas the βCD-based NIP-NSs released ~70%, and the M-βCD-based NIP-NSs released ~50% melatonin for 24 h (Table 6). In all cases, the release profile presented a significant enhancement of the released quantity of melatonin after 23 and 24 h, due to the accumulation of melatonin in the receptor chamber of the Franz Cell during the absence of chamber emptying for a certain period (overnight).

The comparison of MIP-NSs with loaded-NIP-NSs was performed. As it is presented in Figure 7, the amount of released melatonin was higher for the βCD- based NIP-NSs and M-βCD-based NIP-NSs. Within 3 h, the quantity of released melatonin by MIP-NSs and NIP-NSs was approximately the same, followed by an increase for the NIP-NSs, and reaching a plateau for MIP-NSs for 7 h. After 24 h, the released melatonin was significantly increased, particularly for NIP-NSs. The same behavior was observed when the mg of released melatonin from 100 mg of nanosponge (Table 6 and Appendix A) was considered. The loading of melatonin, as a template during the synthesis, improved the interactions of the drug with the polymer structure, and, therefore, the release from MIP-NSs was slower than the NIP-NSs. The percentage of melatonin released from MIP-NSs increased in the first phase of the cream application, whereas it increased from NIP-NSs over the time of 7 h, making NIP-NSs less controllable for a drug delivery system. As already observed in the literature, this could be explained because the NIP-NSs have no specific binding cavities in comparison with MIP-NSs, which, due to the specific network structure, can retain a significant content of the drug [95]. The release kinetics of 1% melatonin mixed with the cream formulation was faster than that of MIP-NSs and NIP-NSs (Figure 8), whereas the release kinetics of 5% mixed with the cream formulation was faster than that of MIP-NSs but more similar to NIP-NSs.

The difference in melatonin loading was dependent on the used monomer to synthesize the nanosponges. The β-CD tended to form a more flexible structure and owned freely accessible cavities to allow the access of drug inside it. On the other hand, M-βCD had a large quantity of methyl groups that did not allow a strong bonding of the melatonin, thus resulting in a high amount of in vitro release during 24 h. The therapeutic efficacy of the drug was dependent on its release, whereas the release from the topical or transdermal dosage form was influenced by various factors such as polymers utilized to load the drug, cream viscosity, etc. The differences in the release kinetics of the formulations of the MIP-NSs and NIP-NSs indicated the different associations within the polymeric structure. The released percentage of melatonin from M-βCD-based MIP-NSs was higher than βCD-based MIP-NSs, even though the melatonin loading into M-βCD-based MIP-NSs was lower. However, the in vitro release results were consistent with the ones of the encapsulation efficiency, because the MIP-NSs (MIP βCD:CA 1:8 20% Melatonin and MIP M-βCD:CA 1:8 20% Melatonin) with the highest encapsulation efficiency (low loading capacity), as previously discussed, presented the lowest drug release percentages. The better-controlled release for MIP-NSs could be explained due to the carboxylic groups, the citric-based MIP-NSs at pH 7.4 (phosphate-buffered solution), which caused the diffusion rate of the buffer into imprinted specific cavities to be slower [96]. The COOH group had a pKa value of 4.6, carrying a negative charge above this value, resulting in ionization, whereas it is neutral and unionized below this value [97]. This could also be related to the low WAC of MIP-NSs (Table 3). As already observed from the literature, the stability of polymers under aqueous conditions is desirable for a controlled release. This indicates the influence of cross-linking in drug release behavior. The immediate therapeutic action can be observed if the drug is directly exposed to the dissolution media [98]. The aim of the polymer as a carrier that releases the drug is to retain and control the concentration of the drug in the blood or target issue. Even though there are several models of release kinetics to describe the drug release from drug-delivery systems, in this study, a solvent-controlled release through the swelling-controlled release was considered [99]. A swelling-controlled system was composed of cyclodextrin-based nanosponges (MIP-NSs and NIP-NSs) that have a three-dimensional cross-linked network structure. These cross-linked polymers were characterized by a great capacity for water absorption (Section 3.4) and, at the same time, a release of the dissolved drug by diffusion through the swollen region [100]. The designing of a controlled drug delivery system requires taking into account several phenomena. The excess amount of soluble drug present within the matrix at specific conditions is considered non-dissolved and thus cannot diffuse. The drug concentration gradient is not increased with the increasing of the initial loading of poorly water-soluble drugs. In these circumstances, the excess of drug within the matrix increases, the absolute amount of drug released within a certain time remains constant, and the relative drug release rate decreases [91]. The findings from the literature can be supported by Table 6, presenting an approximately constant release rate (% and mg of released melatonin) that is not affected by the melatonin concentration (10%, 20%, and 50% melatonin). The slight differences could be related, most likely, to the permeability or thickness of the polymeric membrane [101]. Moreover, the goal of this study was to describe the connection between the drug release kinetics and drug/molecularly and non-molecularly imprinted polymer interaction. The basic release mechanisms of melatonin from MIP-NSs and NIP-NSs were presented, and the models to accurately predict melatonin release profiles needs to be further identified. As a result, the MIP-NSs can be used as a promising formulation for transdermal melatonin application. This in vitro release study will provide insights about the performance of the in vivo studies that are planned in the future.

As already observed in the literature [102], the MIPs could regulate or control the drug release actions by decreasing and increasing the residence time of the drug within the polymer, respectively, to avoid its side effect due to overconcentration of the drug within the body at a particular time. MIPs could help to achieve sustained release because of the affinity of the template to the functional monomer, thereby increasing the residence time of the drug within the body. The findings in this study supported what had already been observed in the previous research about cyclodextrin-based MIP-NSs. The advantage of MIP-NSs over NIP-NSs was presented by Deshmukh et al., who synthesized molecularly imprinted polymers (MIP) of PMDA cross-linked βCD-based NSs. Glucose was used as a template. The synthesized MIP-NSs displayed more significant binding and specificity for glucose, compared to their NIP-NSs [56]. Trotta et al. synthesized MIP-NSs of CDI cross-linked β-CD-based NSs with the L-DOPA as a template. L-DOPA is a non-proteinogenic amino acid widely utilized for the treatment of Parkinson’s disease (PD). Due to the stronger interactions of L-DOPA with the MIP-NSs, it presented a slower release profile than with NIP-NSs. MIP-NSs released ~21% L-DOPA within 4 h, whereas the NIP-NSs released ~77%. This study presented a promising new drug delivery system as an oral formulation for the protection and prolonged release of L-DOPA [57].

Figure 9 describes the schematic structure of the human skin and major routes of transdermal delivery. The stratum corneum, the top layer of the epidermis, causes the permeation barrier properties of human skin [103]. Topical and transdermal delivery approaches have more unique advantages than other delivery routes. The advantages include the topical delivery of a small quantity of a drug to produce a therapeutic effect for skin diseases.

### 3.7. Evaluation of the Pharmaceutical Parameters of Cream Formulations

The formulations underwent a centrifugation test, and the stability of the final optimized cream was observed at room temperature, using the centrifuge, at a certain weight and 4000 RPM for the duration of 30, 60, and 90 min to evaluate syneresis, precipitation, or phase separation. No changes in the texture of the cream or instability phenomena after 30, 60, and 90 min were observed (Figure 10). Figure 11 and Appendix A present the spreadability of the cream formulations. Spreadability was an important property to achieving adequate acceptance and efficacy of the formulations. It had a significant role in the administration of the standard dose of formulation to the skin. The topical cream was considered good if the spreadability decreased because the application was easier [104]. As it can be seen from Appendix A, there were no considerable differences between the raw cream formulation and βCD:CA MIP-NSs (1:8) cream formulation. However, the spreadability of M-βCD:CA MIP-NSs (1:8) cream formulation was the lowest. This can be related to the viscosity of cream formulations (as presented in Appendix A). The M-βCD:CA MIP-NSs 1:8 (10% Melatonin) cream formulation had the highest apparent viscosity. The viscosity is an important property for the processing, handling of the products, and controlling of the drug release since it can affect the therapeutic efficiency of the formulation [105]. The viscosity was higher for M-βCD:CA MIP-NSs 1:8 cream formulation rather than the βCD:CA MIP-NSs 1:8 cream formulation [25,26], an observation that can be related to the swelling capacity and spreadability of MIP-NSs and NIP-NSs (as displayed in Table 3 and Appendix A). Table 3 and Appendix A present the increase of the viscosity of material with the increment of the WAC or swelling capacity. Additionally, the high viscosity of the material caused a low spreadability [106,107]. Another significant factor to determine the efficiency of the cream formulations was the pH. The pH of the prepared formulations must follow the benchmark pH for the skin application (4.5–6). Appendix A presents a more acidic pH (pH = 3–4) than that of the normal pH value of the skin due to citrate-based MIP-NSs, in comparison to the raw cream formulation (pH = 4.90) [26]. However, citric acid is an environmentally friendly, non-toxic, and non-carcinogenic cross-linking agent [108], and therefore it may not appear to have any drawback with the presence in the cream formulation.

Based on this new material proposed for topical melatonin delivery, there is a chance to greatly advance the MIP-NSs development and explore the asset of the transdermal administration over the parenteral and oral routes.

## 4. Conclusions and Perspectives

The green cyclodextrin-based molecularly imprinted nanosponges (MIP-NSs) were successfully synthesized by reacting β-Cyclodextrin (β-CD) or M-β Cyclodextrin (M-β CD), with citric acid as a cross-linking agent at a 1:8 molar ratio in deionized water in the presence of melatonin as a target molecule. In addition, to be compared, β-CD or M-βCD- based non-molecularly imprinted nanosponges (NIP-NSs) were further synthesized and the melatonin, as a target molecule, was loaded (4:6, *v*:*v*, EtOH:H_2_O). The synthesized NIP-NSs were characterized by water absorption capacity (WAC) of ~250%, a fundamental property that affects drug release rates. Moreover, the stability of NIP-NSs was proven by negative zeta potential values (−24 to −33 mV). No degradation of melatonin (storage and thermal stability) was observed during the storage of MIP-NSs at room temperature, and their exposure to light for several weeks. After melatonin loading, an in vitro release study, using a Franz Diffusion Cell, was performed. It presented the ability of MIP-NSs to control and sustain the release of melatonin, supporting a release mechanism in which the percentage of the drug released from the polymer matrix was dependent on the selective interaction between the imprinted cavities and melatonin. Due to the different associations within the polymeric structure, MIP-NSs presented better-controlled drug release performance (pH = 7.4) for 24 h, with 30–50% more released melatonin than NIP-NSs with 50–70% released melatonin. In this study, a cream formulation (oil/water phase) for the delivery of melatonin was introduced. The applicability of this material, mixed with the formulation, was then evaluated for its stability, spreadability, viscosity, and pH. MIP-NSs, due to the presence of binding sites, seemed to be a very promising polymeric device for the controlled and selective release of melatonin. A non-toxic, biodegradable, and biocompatible imprinting material was introduced, and, despite the high-temperature conditions (140 °C and 100 °C) of its synthesis, the inclusion complex formation (nanosponges/melatonin) was not affected. The same inclusion complex was also supported by the adoption of computational methods through static simulations in the gas phase at the GFN2 level.

This study can be used to plan other platforms that can release drugs in a sustained manner. The optimization of the synthesis, characterization, reproducibility, selectivity, recognition, stability, durability, and delivery path of MIP-NSs for a selected template separate to the NIP-NSs is a further research perspective.

## Figures and Tables

**Figure 1 polymers-15-01543-f001:**
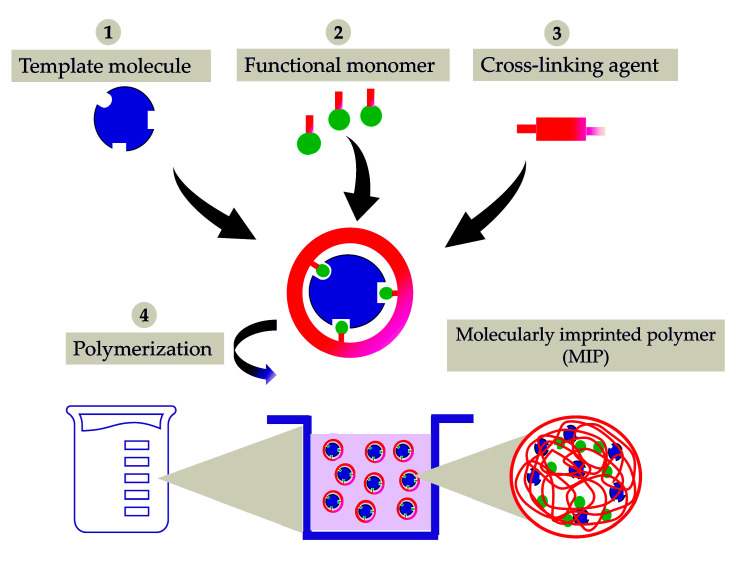
Schematic representation of the synthesis of a molecularly imprinted polymer (MIP) for drug delivery. It is presented as a simple process of molecular imprinting, where the functional monomer, template molecule, and cross-linking agent are mixed to produce a highly cross-linked polymer, under certain reaction conditions.

**Figure 2 polymers-15-01543-f002:**
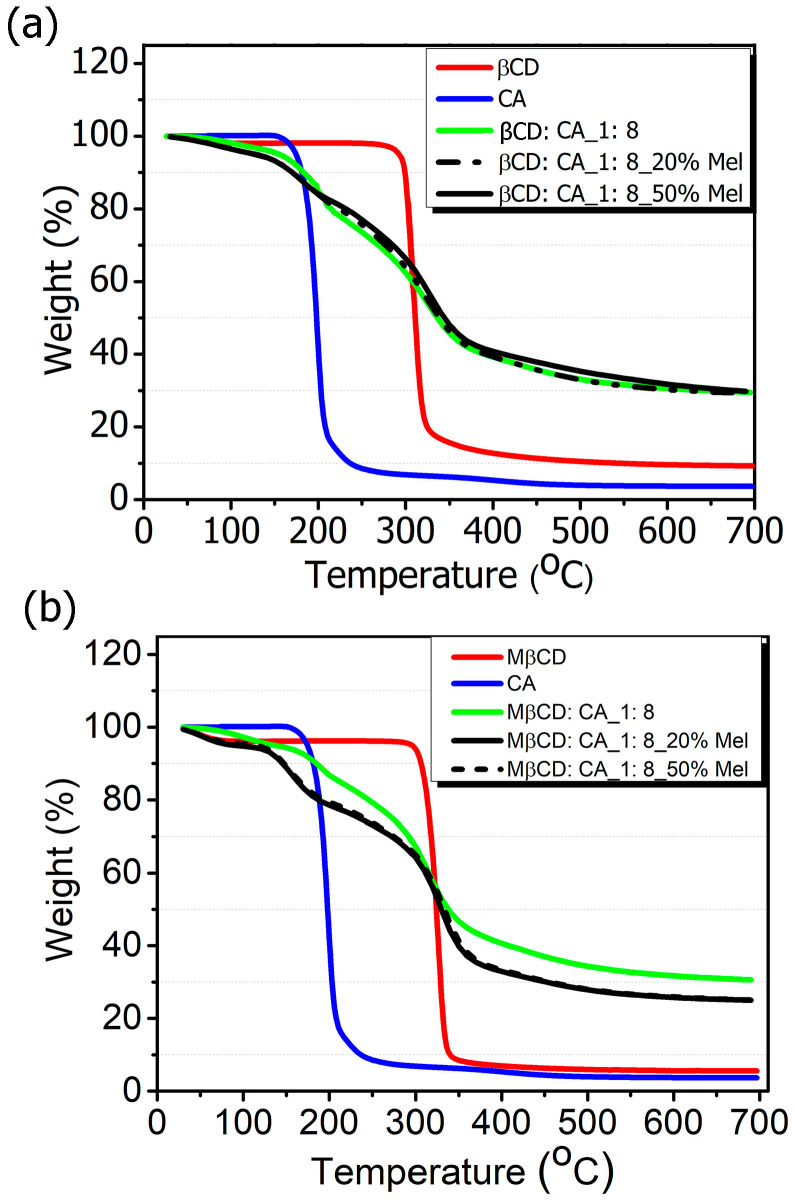
Thermogravimetric analysis (TGA) of (**a**) βCD-based MIP-NSs and βCD-based NIP-NSs and (**b**) M-βCD-based MIP-NSs and M-βCD-based NIP-NSs.

**Figure 3 polymers-15-01543-f003:**
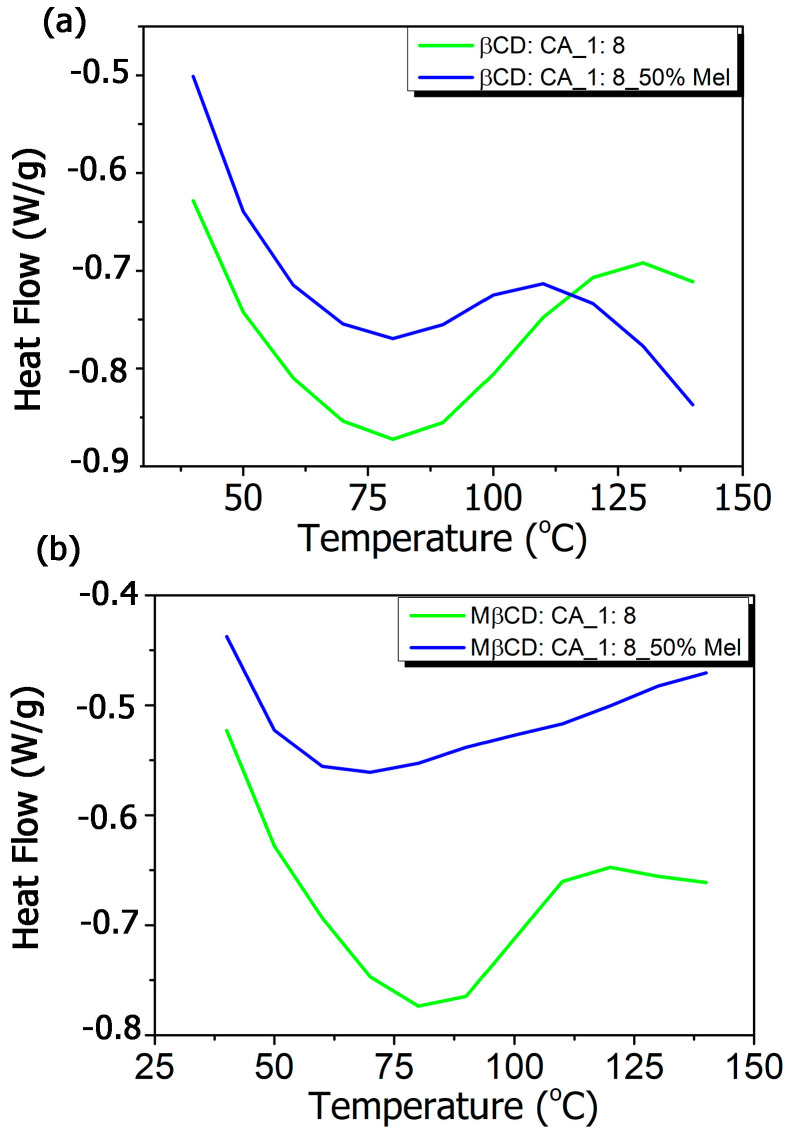
Differential scanning calorimetry (DSC) thermograms of (**a**) βCD-based MIP-NSs and βCD-based NIP-NSs, and (**b**) M-βCD-based MIP-NSs and M-βCD-based NIP-NSs.

**Figure 4 polymers-15-01543-f004:**
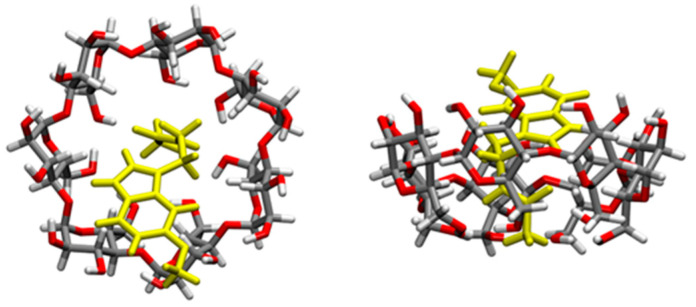
Structures of the inclusion complex between melatonin and β-CD in the gas phase, of which the IR vibrational frequencies are computed. Melatonin is depicted in yellow, while atoms in β-CD grey (C), red (O), and white (H).

**Figure 5 polymers-15-01543-f005:**
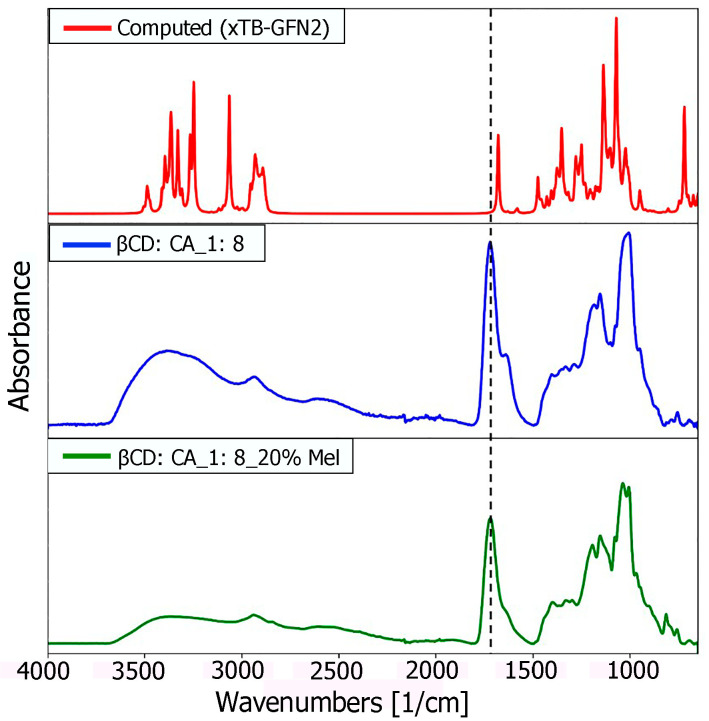
Comparison of computational (red), and experimental (blue and green) spectra of the inclusion complex between melatonin and β-CD in the gas phase.

**Figure 6 polymers-15-01543-f006:**
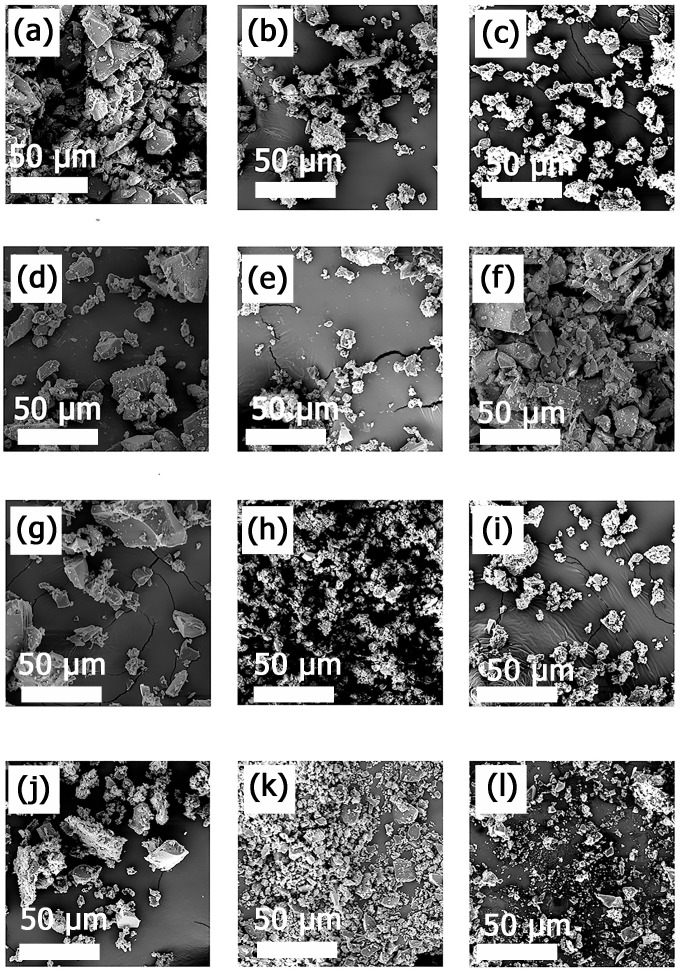
SEM images of (**a**) NIP βCD:CA 1:8, (**b**) MIP βCD:CA 1:8 3% Melatonin, (**c**) MIP βCD:CA 1:8 5% Melatonin, (**d**) MIP βCD:CA 1:8 10% Melatonin, (**e**) MIP βCD:CA 1:8 20% Melatonin, (**f**) MIP βCD:CA 1:8 50% Melatonin; (**g**) NIP M βCD:CA 1:8, (**h**) MIP M βCD:CA 1:8 3% Melatonin, (**i**) MIP M-βCD:CA 1:8 5% Melatonin, (**j**) MIP M βCD:CA 1:8 10% Melatonin, (**k**) MIP M βCD:CA 1:8 20% Melatonin, (**l**) MIP M βCD:CA 1:8 50% Melatonin. The magnification is 1.5 k×, and the scale bar is 50 μm.

**Figure 7 polymers-15-01543-f007:**
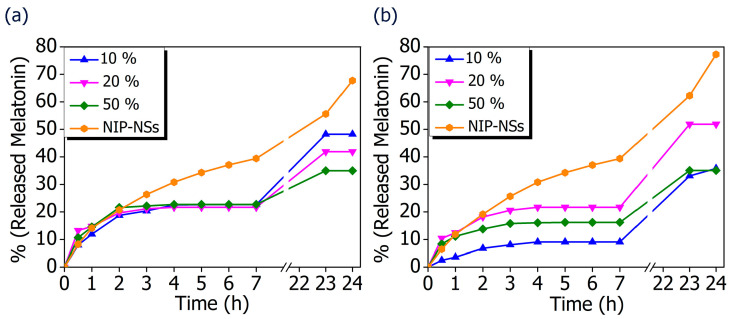
In vitro release curves of melatonin loaded into MIP-NSs (10%, 20%, and 50% Melatonin), and NIP-NSs (20% Melatonin). (**a**) βCD-based NIP-NSs and βCD-based MIP-NSs; and (**b**) M-βCD-based NIP-NSs and M-βCD-based MIP-NSs.

**Figure 8 polymers-15-01543-f008:**
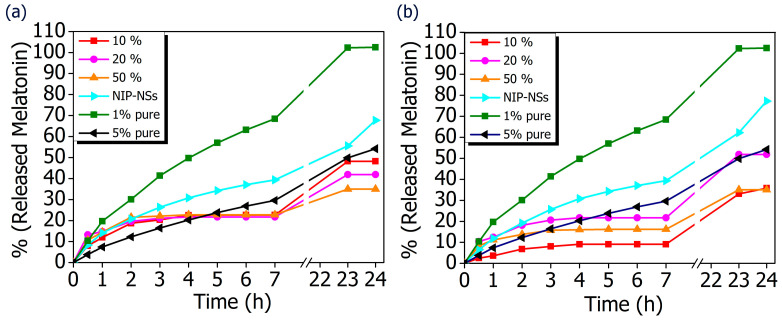
In vitro release curves of cream formulation with melatonin loaded into MIP-NSs (10%, 20%, and 50% Melatonin), and NIP-NSs (20% Melatonin). (**a**) βCD-based NIP-NSs or βCD-based MIP-NSs; and (**b**) M-βCD-based NIP-NSs or M-βCD-based MIP-NSs. The results are compared with the cream formulation that only contains 1% and 3% melatonin.

**Figure 9 polymers-15-01543-f009:**
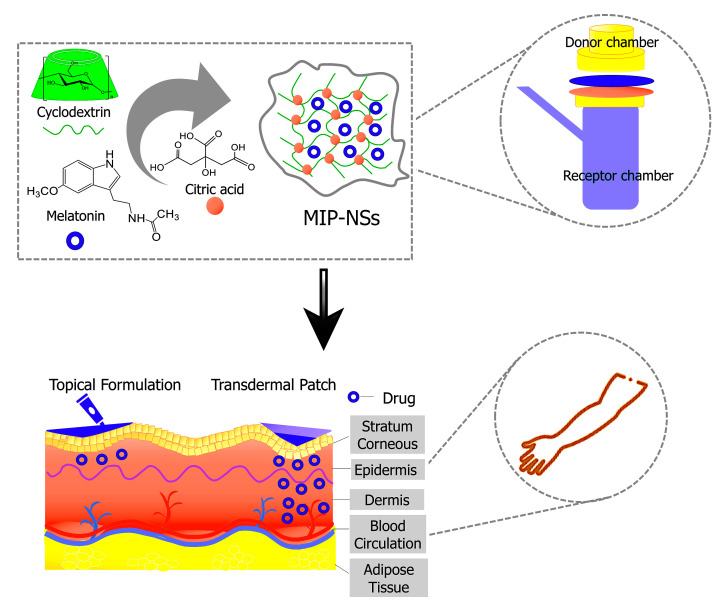
Schematic representation of MIP-NSs proposed as a new strategy for topical and transdermal melatonin administration.

**Figure 10 polymers-15-01543-f010:**
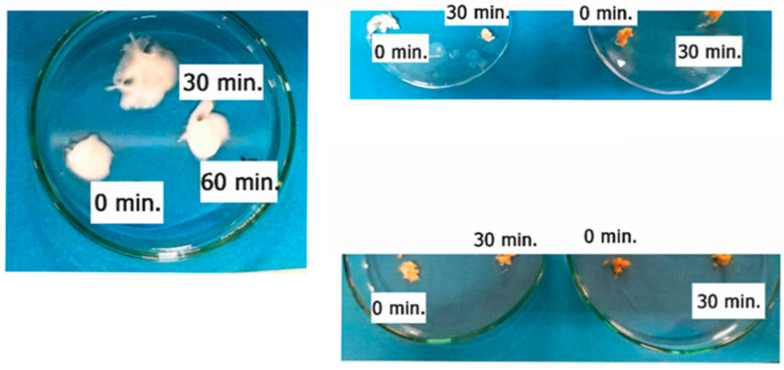
The centrifugation test of the cream formulations.

**Figure 11 polymers-15-01543-f011:**
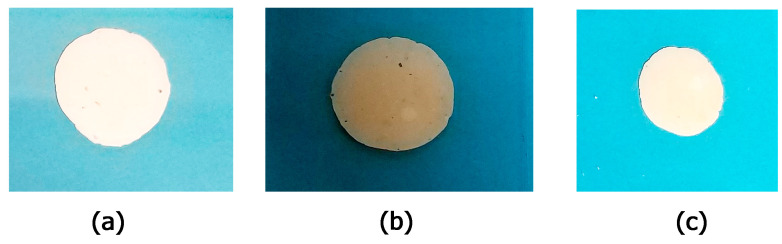
The spreadability of the (**a**) cream formulations; (**b**) M-βCD:CA 1:8 50% Melatonin cream formulation; (**c**) βCD:CA 1:8 50% Melatonin cream formulation.

**Table 1 polymers-15-01543-t001:** The amounts of reactants to synthesize cyclodextrin-based molecularly non-imprinted and imprinted nanosponges (NIP-NSs and MIP-NSs). The numbers 1:8 indicate the molar ratio of β-Cyclodextrin/Methyl-βcyclodextrin: citric acid (β-CD/M-βCD:CA).

Samples	β-CD/M-βCD (mmol)	CA (mmol)	SHP (mmol)	Melatonin (mmol)
NIP β-CD:CA 1:8	4.41	35.23	8.58	-
NIP M-βCD:CA 1:8	4.22	33.78	8.58	-
MIP β-CD:CA 1:8/10% Melatonin	4.41	35.23	8.58	0.43
MIP β-CD:CA 1:8/20% Melatonin	4.41	35.23	8.58	0.86
MIP β-CD:CA 1:8/50% Melatonin	4.41	35.23	8.58	2.19
MIP M-βCD:CA 1:8/10% Melatonin	4.22	33.78	8.58	0.38
MIP M-βCD:CA 1:8/20% Melatonin	4.22	33.78	8.58	0.82
MIP M-βCD:CA 1:8/50% Melatonin	4.22	33.78	8.58	2.11

**Table 2 polymers-15-01543-t002:** Composition of the cream mixtures.

Ingredient	Quantity
Aqueous phase	
Glycerol	3.00 g
H_2_O	40.00 mL
Oil phase	
Almond oil	3.00 g
Stearate PEG-100	5.00 g
Beeswax	4.00 g
Cetyl alcohol	4.00 g
BHT	0.50 g

**Table 3 polymers-15-01543-t003:** Water absorption capacity (WAC) of NIP-NSs and MIP-NSs.

Samples	WAC (%)	Zeta Potential (mV)
NIP βCD:CA 1:8	126 ± 5	−33.20 ± 1.08
NIP M-βCD: CA 1:8	237 ± 18	−24.20 ± 0.53
MIP βCD:CA 1:8/20% Melatonin	121 ± 10	-
MIP M-βCD:CA 1:8/20% Melatonin	53 ± 6	-

**Table 4 polymers-15-01543-t004:** The loading capacity and encapsulation efficiency of MIP-NSs and melatonin loaded-NIP-NSs.

Samples	HPLC-UV	
	LC (%)	EE (%)
NIP βCD: CA_1: 8_20% Melatonin	6.51	39.11
NIP M-βCD: CA_1: 8_20% Melatonin	4.37	26.21
MIP βCD: CA_1: 8/20% Melatonin	1.39	88.13
MIP M-βCD: CA_1: 8/20% Melatonin	1.01	64.59

**Table 5 polymers-15-01543-t005:** The CHNS-O elemental analysis, loading capacity, and encapsulation efficiency of MIP-NSs.

Samples	N (%), CHNS-O	HPLC-UV	
		LC (%)	EE (%)
MIP βCD:CA_1:8/10% Melatonin	<0.10	0.57	72.13
MIP βCD:CA_1:8/20% Melatonin	0.22	1.39	88.13
MIP βCD:CA_1:8/50% Melatonin	0.52	3.87	99.80
MIP M-βCD:CA_1:8/10% Melatonin	<0.10	0.36	46.27
MIP M-βCD:CA_1:8/20% Melatonin	0.21	1.01	64.59
MIP M-βCD:CA_1:8/50% Melatonin	0.53	2.84	74.90

**Table 6 polymers-15-01543-t006:** % and mg (mg/100 mg of NSs) of released melatonin from the cream formulations with βCD-based NIP-NSs (20% Melatonin), M-βCD-based NIP-NSs (20% Melatonin), βCD-based MIP-NSs (10%, 20%, and 50% Melatonin), M-βCD-based MIP-NSs (10%, 20%, and 50% Melatonin), 1% melatonin, and 5% melatonin.

Samples	% Release	mg/100 mg NSs
NIP βCD:CA_1:8/20% Melatonin	68%	0.33 mg
MIP βCD:CA_1:8/10% Melatonin	48%	0.25 mg
MIP βCD:CA_1:8/20% Melatonin	42%	0.24 mg
MIP βCD:CA_1:8/50% Melatonin	35%	0.23 mg
NIP M-βCD:CA_1:8/20% Melatonin	77%	0.35 mg
MIP M-βCD:CA_1:8/10% Melatonin	36%	0.13 mg
MIP M-βCD:CA_1:8/20% Melatonin	52%	0.25 mg
MIP M-βCD:CA_1:8/50% Melatonin	35%	0.18 mg
1% Melatonin	~100%	1.10 mg
5% Melatonin	54.21%	2.71 mg

## Data Availability

Not applicable.

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
