# Peer review of "A Comparison between the Molecularly Imprinted and Non-Molecularly Imprinted Cyclodextrin-Based Nanosponges for the Transdermal Delivery of Melatonin"

_polymers, 2023, doi:10.3390/polym15061543_

Round 1
Reviewer 1 Report
Comments;
1. The abstract must show the important results. please rewrite it by including the important results.
2. The melatonin sustained release formulations are developed for oral, transdermal, intranasal, and transmucosal administrations. if so why this study is needed. why it should be considered novel.
3. Please write more detail about the SEM morphology. it is not very clear. please use arrow to show the changes in the SEM images of the different samples.
4. melatonin is partially water-soluble, and therefore leads to insufficient absorption, then how did it was loaded, please make it easily readable and unsderstandable.
5. Please elaborate the main mechanism of drug release.
Reviewer 2 Report
The present article entitled “A Comparison between the Molecularly Imprinted and Non-Molecularly Imprinted Cyclodextrin-based Nanosponges for the Transdermal Delivery of Melatonin" describes the cyclodextrin (CD)-based molecularly imprinted nanosponges (MIP-NSs) with melatonin as a template molecule. Thus, this reviewer recommends the publication of this work in this Journal after addressing the following concerns.
Comments
1. Quantitative information should be reflected in the abstract.
2. In the introduction section should be provided the more references.
3. The author should provide the citation in the section 2.2.1. Synthesis of cyclodextrin-based molecularly imprinted and non-molecularly imprinted nanosponges (MIP-NSs and NIP-NSs).
4. 2.2.16. In vitro release study; in vitro should be italic.
5. The conclusion should be concise and revised with the outstanding point of this work.
6. Typographical errors and superfluous spaces throughout the manuscript should be corrected.
Reviewer 3 Report
The paper is suggested for publication after minor improvement:
1. How about the loaded amout of Melatonin on the polymer per g?
2. Comparison can be further strengthened with current studies in section 3.5-3.6.
3. Porosity of nanosponges should be discovered, e.g. specific surface area, porosity, pore size, etc.
4. The formulas and quantitative methods should be supported by references.
5.How about the composition of cream formulation on drug release? was it determined by optimization?
6. Where is the discussion on section 2.2.14?
7. Zeta potential of the nanosponges is suggested for measurement.
8. How about the molar ratio of melatonin and β-cyclodextrin in their complex through experimental validation? also 1:1?
9. Can the nanosponges be reused?
Reviewer 4 Report
The paper "polymers-2233038" describe the synthesis, characterization and development of cream formulations of molecular Imprinted and non-molecular imprinted cyclodextrin-based nanosponges for the transdermal delivery of melatonin. The study is interesting and may be published after minor revision.
Some suggestions of improvement are given:
1.Try to better describe the molecular interactions of CD-melatonin. Give the binding energy and its terms as obtained by molecular mechanics. One may predict easily the contribution of electrostatic or vdw terms from docking studies. see please: Sakellaropoulou, Aikaterini, Angeliki Siamidi, and Marilena Vlachou. 2022. "Melatonin/Cyclodextrin Inclusion Complexes: A Review" Molecules 27, no. 2: 445. https://doi.org/10.3390/molecules27020445
Li, Hui, Guolei Zhang, Wei Wang, Changbao Chen, Lili Jiao, and Wei Wu. 2021. "Preparation, Characterization, and Bioavailability of Host-Guest Inclusion Complex of Ginsenoside Re with Gamma-Cyclodextrin" Molecules 26, no. 23: 7227. https://doi.org/10.3390/molecules26237227
2.Compare your results with previous studies.
3.increase figures 3, 8, S1, S2, and S8 resolution, the text is almost illegible.
4.revise English and typos: line 4 at section 227 "carbon type" should be "carbon tape"
5. Decreases y axis scale from 95 to 105 % to better see the differences in Figure S3.
6. Figure S4 has no x axis units.
Round 2
Reviewer 1 Report
The authors did well. No more comments